# Experimental Study on Corrosion Performance of Oil Tubing Steel in HPHT Flowing Media Containing O_2_ and CO_2_

**DOI:** 10.3390/ma13225214

**Published:** 2020-11-18

**Authors:** Yihua Dou, Zhen Li, Jiarui Cheng, Yafei Zhang

**Affiliations:** School of Mechanical Engineering, Xi’an Shiyou University, Xi’an 710065, China; xsyoucjr@163.com (Y.D.); lizhenxsyu@sina.com (Z.L.); effyzhang@126.com (Y.Z.)

**Keywords:** high pressure and high temperature, O_2_–CO_2_ coexistence environment, flow-induced corrosion

## Abstract

The high pressure and high temperature (HPHT) flow solution containing various gases and Cl^−^ ions is one of the corrosive environments in the use of oilfield tubing and casing. The changing external environment and complex reaction processes are the main factors restricting research into this type of corrosion. To study the corrosion mechanism in the coexistence of O_2_ and CO_2_ in a flowing medium, a HPHT flow experiment was used to simulate the corrosion process of N80 steel in a complex downhole environment. After the test, the material corrosion rate, surface morphology, micromorphology, and corrosion product composition were tested. Results showed that corrosion of tubing material in a coexisting environment was significantly affected by temperature and gas concentration. The addition of O_2_ changes the structure of the original CO_2_ corrosion product and the corrosion process, thereby affecting the corrosion law, especially at high temperatures. Meanwhile, the flowing boundary layer and temperature changed the gas concentration near the wall, which changed the corrosion priority and intermediate products on the metal surface. These high temperature corrosion conclusions can provide references for the anticorrosion construction work of downhole pipe strings.

## 1. Introduction

The thermal recovery technology of heavy oil, which is applied to the production of tight and shale oil in the oil field, has the characteristics of high construction temperature, complex flowing medium, and strong corrosiveness. Generally, the production process contains a variety of anions and cations and multicomponent gases, with temperature higher than 150 °C. As the only channel for oil or gas exploitation, the downhole tubing string is inevitably corroded by the high pressure, high temperature, and complex flowing solution, which cause deformation and fracture damage of tubing string, thereby affecting well site safety [1].

N80 tubing steel, which is commonly used in thermal recovery wells, is damaged by the corrosion of O_2_, CO_2_, and H_2_S in the liquid containing HCO_3_^−^, Cl^−^, SO_4_^2−^, Ca^2+^, Mg^2+^, and Na^+^ [2]. The corrosion in the high temperature flowing liquid environment where O_2_, CO_2_, and Cl^−^ coexist is the focus of intense research for N80 tubing steel. The essence of the CO_2_ corrosion for tubing steel is the electrochemical corrosion of metals in aqueous carbonate solution [3]. Thus, many studies have investigated CO_2_ corrosion based on these factors. Bai et al. [4] investigated the influence of CO_2_ partial pressure on the properties of J55 carbon steel in 30% crude oil/brine at 65 °C. The corrosion rate significantly increases as the CO_2_ partial pressure increases from 0 to 1.5 MPa and decreases from 1.5 to 5 MPa, and the surface of J55 carbon steel is covered by FeCO_3_ and CaCO_3_. The CO_2_ partial pressure changes the system pH and CO_2_ solubility in crude oil, which further affects the formation and protection performance of the corrosive film. Zhang et al. [5] studied the corrosion behavior of N80 carbon steel under dynamic supercritical CO_2_ (8 MPa)–water environment. The results demonstrated no essential difference in the electrochemical corrosion mechanism between supercritical CO_2_ and nonsupercritical CO_2_ (5 MPa) environments. The corrosion rate under dynamic condition is higher than that under static condition in the initial time and flowing fluid hinders the formation of corrosive film and then increases the corrosion rate.

De Waard et al. [6] studied the reaction process of the anode and cathode through the point-position mechanical polarization curve, and the anode reaction formula (Fe → Fe^2+^ + 2e^−^) containing OH^−^ was obtained. Nesic et al. [7] divided the pH value of the solution into PH < 4, 4 < PH < 6, and PH > 6; for the CO_2_ anode electrochemical corrosion reaction process, an intermediate product Fe(CO_3_)OH might be generated. Linter et al. [3] reported that Fe(OH)_2_ occurs first in the process of CO_2_ corrosion, and then FeCO_3_ would be further generated. The CO_2_ corrosion rate is generally controlled by the reaction of the cathode. Scholars locally and internationally have conducted many studies on cathode corrosion. The results showed that the corrosion rate of steel in CO_2_ solution is controlled by H evolution kinetics [8].

O_2_ is a catalyst for CO_2_ corrosion and is also a cathodic depolarizer in the corrosion process of steel. When the protective film is not formed on the carbon steel surface, the corrosion rate will increase with the O_2_ content, while the CO_2_ content plays a decisive role in the corrosion of carbon steel after the O_2_ content reaches saturation in solution. Once the protective film is formed on the surface, the corrosion of carbon steel is not affected by the O_2_ content [9]. In the presence of O_2_, a double-layer film structure is present in the CO_2_ corrosive system, the inner of which is mainly Fe_2_O_3_, and the outer is the loose and porous corrosion product FeCO_3_ [10]. The increase in O_2_ content will result in a dense outer layer of corrosion product film to protect the steel surface. Another study [11] has found that the corrosion products on the 3Cr steel surface are formed by the accumulation of granular products under high temperature conditions where O_2_ and CO_2_ coexist, and the outer corrosion scales are mainly FeCO_3_ and Fe oxides.

The influence of Cl^−^ is reflected in two aspects, as follows [12]: (1) the possibility of passivation film formation on the sample surface is reduced, or the damage of the passivation film is accelerated; therefore, the local corrosion reaction is promoted. (2) The solubility of CO_2_ in the aqueous solution is reduced, which could alleviate the corrosion of carbon steel. The corrosion product film on the surface of N80 steel is denser, and the adhesion is high at a low Cl^−^ content (5000 mg/L), which indicates that corrosion resistance is improved [13]. When the Cl^−^ content increases twofold, the protective effect of the corrosive film decreases due to the weak compactness of the corrosive film, thereby resulting in an accelerated corrosion rate of steel. Some studies have compared the effects of Cl^−^ concentration and CO_2_ concentration on the corrosion of carbon steel. The test results of Wang et al. [14] showed that the corrosion rate of N80 steel in 1–3 wt% NaCl solution is the highest. Meanwhile, there is a critical partial pressure of CO_2_ in the chloride-containing solution that affects the corrosion rate. Zhang et al. [15] tested the corrosion law of N80 steel under different CO_2_ partial pressures in detail. The results show that the increase in CO_2_ partial pressure accelerates the corrosion reaction but, at the same, time forms a dense protective layer. However, in a chloride-containing solution, Cl^−^ is sufficient to penetrate the protective layer and react with the base metal [16].

High-pressure and high-temperature (HPHT) multicomponent media are the most common corrosion environments for N80 steel underground. To study the corrosion law of pipes, we used a high-temperature and high-pressure reactor to create an HPHT flow solution to simulate the corrosion environment. The corrosion rate, surface morphology, surface composition, and corrosion layer profile morphology results were obtained in the test. Meanwhile, the N80 corrosion process in an environment of coexisting CO_2_ and O_2_ has been discussed in various ways.

## 2. Experimental

### 2.1. Experimental System

The HPHT corrosion system was used for the flow-induced corrosion test of the N80 steel, including the HPHT autoclave (Weihai Global Chemical Machinery MFG Co., Ltd., Weihai, China), the lifting platform, the control cabinet, the booster pump, and the air source. As shown in Figure 1, the inside of the autoclave was made of Hastelloy C-276, which can resist corrosion under most conditions. The exterior of the autoclave is made of 304 stainless steel. The samples were installed on the fixture and rotated with the agitator shaft and fixture. Approximately 5 L of mixed fluid (simulated produced fluid in oilfield) was injected into an autoclave and stirred by an agitator. The dissolved oxygen was purged in the solution with injected nitrogen gas for 4 h under a pressure of 0.5 MPa. The autoclave was pressured with pure N_2_ to the experimental values (total pressure value and CO_2_ partial pressure) with CO_2_ gas to the experimental values for 72 h at the flow velocity of 1 m/s. The surface microstructure of the corrosion product scales on the surface of the corroded samples was analyzed by JSM-6390 SEM (JEOL, Tokyo, Japan). The composition of the corroded samples was performed by XRD (LabX XRD-6000, SHIMADZU Co., Ltd., Kyoto, Japan).

### 2.2. Experimental Conditions

The composition of the experimental material, N80 tubing steel, is listed in Table 1. The samples were machined to dimensions of 50 × 13 × 1.5 mm. Before the experiment, the samples were placed in acetone to remove the surface oil and sequentially polished with 500#, 800#, 1200#, and 2000# sandpaper. The samples were scrubbed two times with distilled cotton in distilled water and cleaned for 15 s with distilled water. The samples were dried in cold air and placed in the dryer for 24 h. Finally, the samples were weighed using a digital balance with an accuracy of 0.1 mg.

After the experiment, the corroded samples were photographed to record the surface variations and then immersed in an acid solution (500 mL of HCl and 3.5 g hexamethylenamine diluted with water to 1000 mL) for 10 min to remove product films [4]. The samples were placed on absolute ethanol for washing two times until the acid cleaning solution on the surface was completely removed. The samples were dried and weighed before the test. When testing XRD, the product film of the sample was peeled off, and the crystal grain was ground to 0.1–10 μm. These grains were filled into the material tray and pressed to make the surface flat.

The volume of the produced liquid used in each set of experiments was 5 L, and the liquid ingredients are listed in Table 2. The total pressure of HPHT corrosion was 5 MPa. The experimental temperature varied from 50 to 200 °C, and the solution flow velocity was fixed at 1 m/s. The CO_2_ partial pressures were 0.25, 0.50, and 0.75 MPa, and the O_2_ partial pressures were adjusted to 0.05, 0.10, and 0.15 MPa, respectively.

## 3. Results

### 3.1. Effects of Solution Temperature and CO_2_ Partial Pressure on Corrosion Rate

The corrosion rate of N80 steel measured by changing the CO_2_ concentration and temperature in the CO_2_–O_2_ coexisting environment (0.25 MPa CO_2_ and 0.05 MPa O_2_) is shown in Figure 2and Figure 3. The results show that the corrosion rating of N80 steel in this condition is severe or extremely severe according to the NACE SP 0775-2013 standard. When the partial pressure of CO_2_ is 0.25 MPa, the corrosion rate increases with increasing temperature. When the partial pressure of CO_2_ is greater than 0.5 MPa, the corrosion rate will have a maximum value at 100 °C with the increase in temperature. A comparison of the corrosion rate at different CO_2_ partial pressures is shown in Figure 3. As the temperature increases, corrosion rate varies directly with CO_2_ partial pressure at a temperature less than 100 °C and inversely at a temperature higher than 100 °C. This result indicates that the increase in CO_2_ concentration at low temperature promotes the corrosion reaction, whereas the change in CO_2_ concentration at high temperature weakens the effect of corrosion.

### 3.2. Macroscopic Morphology and Product Components

Figure 4 shows the surface morphology of N80 steel after corrosion at different temperatures in an O_2_ and CO_2_-containing environment. The results show that thick corrosion products are attached to the surface of the corroded material, which causes the metal matrix to be completely covered by precipitated scales. The product films on the surface are loose and thick, and a portion of them falls off at 50 and 100 °C. As the temperature increases from 100 to 200 °C, the product films become thin and smooth, and the color changes from deep red to black. The morphology of N80 steel after corrosion in a high-temperature environment containing 0.05 MPa O_2_ by changing the partial pressure of CO_2_ is shown in Figure 5. When the partial pressure of CO_2_ is 0.25 MPa, the dark-brown corrosion surface is relatively flat, and a loose corrosion product film is attached. No hard scale is present on the surface of the corroded sample because the solubility of CO_2_ will decrease at 200 °C. When the partial pressure of CO_2_ reaches 0.75 MPa, the surface oxide of the material increases, and the product film thickens.

The SEM morphology and XRD test results of corrosion products of N80 steel in an environment of coexisting CO_2_–O_2_ at different temperatures are shown in Figure 6. The results show that the corrosion product film thickness is mainly composed of FeCO_3_ and Fe_2_O_3_ at 50 °C. The granular products on the surface are wrapped in the corrosion layer at 100 °C, which causes the surface of the sample to be uneven. The corrosion products are mainly composed of FeCO_3_, Fe_2_O_3_, and Fe_3_O_4_. Fe_2_O_3_ is oxidized again to Fe_3_O_4_ at 150 °C, which increases the particle attachment. When the temperature continues to increase to 200 °C, the product structure on the surface of the material is stable and dense.

The SEM morphology and XRD test results of corrosion products of N80 steel at different CO_2_ partial pressures are shown in Figure 7. The SEM result with a CO_2_ partial pressure of 0.25 MPa shows that the corrosion products on the surface layer are relatively thin and have not been connected to form an overall structure. When the partial pressure of CO_2_ increases to 0.5 MPa, the product film falls off significantly, and large corrosion cracks appear on the surface. The iron oxides completely cover the original corrosion products to form a double-layer film structure on the surface in a high-concentration CO_2_ environment (P_CO2_ = 0.75 MPa). This kind of membrane structure is not dense or unstable and is easily washed off by the solution, which causes the internal matrix to continue to corrode. The XRD measurement results of the corrosion product of the material show that the surface of the corroded material is covered with FeCO_3_ grains and iron oxides because when the corrosive environment contains CO_2_ and O_2_, FeCO_3_ and Fe(OH)_2_ corrosion product films will appear on the initial surface, and then O_2_ in the environment will oxidize with Fe(OH)_2_ and FeCO_3_ to form iron oxides [3,7]. The corrosion products on the final surface are mainly iron oxides, such as FeCO_3_, Fe_2_O_3_, and Fe_3_O_4_. The outer layer of the formed double layer corrosion product film is mainly composed of FeCO3 and Fe oxides, while the inner layer is mainly composed of regular grained FeCO_3_. Compared with a single CO_2_ corrosion product, the appearance of oxides reduces the integrity of the corrosion product film and increases the corrosion rate of the material.

Figure 8 shows the surface structure after removing the outer layer of the attached product film. The corrosion surface is composed of small corrosion pits that are connected to uneven honeycomb areas at 50 °C. At 100 and 150 °C, the corroded surface is spliced by a large area of massive corrosion pits. The original corrosion products are composed of flaky structures due to the gradual expansion of the local fine pits to form a certain area, and the corrosion products gradually accumulate to form a layer. Under the action of liquid shearing force, the flaky corrosion products on the surface of the material fall off to form a large uneven area. Deep corrosion pits appear on the film surface at 150 °C, and a small amount of corrosion product is mixed in the corrosion pits. This result indicates that the flowing liquid penetrated into the material matrix through the cracks, which causes serious pitting corrosion in the internal surface. The surface of the material is relatively regular and smooth without evident pitting after corrosion at 200 °C, and the corrosion products are mainly flakes.

Figure 9 shows the cross-sectional morphology of the corrosion sample. According to different positions, the sample surface can be divided into an adhesion layer, corrosion layer, and matrix. Among these layers, the adhesion layer is the loosest and consists of crystalline particles or oxide. The corrosion layer is mainly formed by the accumulation of reaction intermediate products and dense corrosion product. The corrosion layer is not enough to prevent macromolecules and ions from contacting the metal surface, especially in flowing liquids, because it contains large pores. For the same reason, the corrosion layer is a porous medium, which cannot prevent Cl ions from contacting the substrate. Hence, when the temperature is in the range of 100 to 150 °C, serious continuous corrosion pits appear in the corrosion layer and extend to the depths of the matrix.

## 4. Discussion

Given that the total pressure in this experiment is 5 MPa, according to the law of O_2_ and CO_2_ gas saturation pressure changing with temperature, the fluid in the kettle is in the form of gas–liquid coexistence at this experimental temperature. During the rotation of the sample, the surface of the material is subjected to the fluid shear force, which accelerates the rupture of the corrosion product film. Meanwhile, the rapid fluid flow accelerates the convective mass transfer of the gas in the boundary layer of the sample surface, thereby changing the corrosion process. According to the characteristics shown by the experimental results, the main factor affecting the corrosion of N80 in the high-temperature and high-pressure flowing media is the change in the reaction of O_2_ or CO_2_ under different flow velocities and temperatures.

### 4.1. Formation of Corrosive Environment Containing CO_2_

The essence of the corrosion of CO_2_ and steel materials is the electrochemical reaction of metals in carbonic acid solution. Although the carbonic acid produced by dissolving CO_2_ in water is a dibasic acid, the acidity of carbonic acid is higher than that of hydrochloric acid under the same pH value, which will cause serious corrosion to steel materials. Nesic [7] divided the pH value of the solution as follows: pH < 4, 4 < pH < 6, and pH > 6. Nesic believes that Fe(CO_3_)OH intermediate products can be generated during the electrochemical reaction of the anode. Thus, the anode reaction is as follows:
(1)$$pH<4\;H_{3}O^{+}+e^{-}\to H+H_{2}O,$$
(2)$$4<pH<6\;H_{2}{\mathrm{CO}}_{3}+e^{-}\to H+{\mathrm{HCO}}_{3}^{-},$$
(3)$$pH>6\;{2\mathrm{HCO}}_{3}^{-}+2e^{-}\to {2H}_{2}{+2\mathrm{CO}}_{3}^{2-},$$

Moreover, the overall reaction equation is as follows:(4)$${\mathrm{Fe}+\mathrm{CO}}_{2}{+H}_{2}O\to {\mathrm{FeCO}}_{3}{+H}_{2}$$

In this experiment, the pH of several groups of media is between 4 and 6. As shown in Figure 10, the CO_2_ corrosion rate decreases with increasing temperature. Therefore, when the temperature is less than 100 °C, a H depolarization process exists in the electrochemical reaction. The process is completed by HCO_3_^−^ and H^+^, which are decomposed by H_2_CO_3_ in the solution. The higher the partial pressure of CO_2_ is, the higher the solubility of H_2_CO_3_ is, and the higher the H^+^ concentration produced by the hydrolysis of H_2_CO_3_ in the solution will be, thereby resulting in fast corrosion of the material [17,18]. However, in a high-temperature environment (T > 150 °C), the decrease in the solubility of CO_2_ and increase in pH of the solution reduce the anode reaction rate. Therefore, the corrosion rate increases with the increase in CO_2_ concentration at low temperature but decreases at high temperature.

### 4.2. Diffusion and Reaction of O_2_ in CO_2_ Solution

The results of comparing the corrosion rate of N80 steel with and without O_2_ are shown in Figure 11. In an environment containing only CO_2_ gas, the surface corrosion rate of N80 shows a law that first increases and then decreases. When the temperature reaches 200 °C, the corrosion rate decreases to <0.5 mm/a. In an environment containing only O_2_ gas, the corrosion rate increases with the increase in temperature, and the maximum corrosion rate occurs at 200 °C. When O_2_ is added to the CO_2_ corrosive medium, the corrosion rate is higher than that of a single gas environment. The corrosion ratio obtained by comparison is shown in Figure 12. When the CO_2_ partial pressure is less than 0.5 MPa, the corrosion rate change law is similar to that of a single CO_2_ environment. However, when the partial pressure of CO_2_ is 0.75 MPa, the corrosion rate continues to increase with the increase in temperature, which indicates that the addition of O_2_ significantly changes the CO_2_ corrosion process in a high-concentration and high-temperature environment. Given that the flow velocity does not change, the forced convection flow in the boundary layer of the same phase medium is close. The results show that the corrosion contrast at different temperatures is significantly different, which indicates that the O_2_ mass transfer law in the corrosion product film affects the electrochemical reaction of the substrate.

The diffusion and mass transfer of O_2_ in the film restricts the rate of the depolarization reaction because the N80 surface forms a FeCO_3_ film on the metal surface in the H_2_CO_3_ solution. In a steady-state mass transfer unit, dC/dt = 0, the total molar flux is defined by the Nernst–Plank equation, as follows [18]:(5)$$J=-D\frac{\mathrm{dC}}{\mathrm{dx}}-\frac{\mathrm{nFDC}}{\mathrm{RT}}\frac{\mathrm{df}}{\mathrm{dx}}{+\mathrm{Cv}=J}_{d}{+J}_{m}{+J}_{c}.$$

Take the corrosion of N80 steel in water at 150 °C and 5 MPa pressure as an example. According to the first boundary condition of Fick’s second law [19], the expression of reactant concentration at any position in the boundary layer is obtained as follows:(6)$$\frac{C^{y}{-C}^{a}}{C^{w}{-C}^{a}}=1-\mathrm{erf}(\frac{y}{\sqrt{4\mathrm{Dt}}}),$$
(7)$$C^{y}{=C}^{w}{-(C}^{w}{-C}^{a})\mathrm{erf}(\frac{y}{\sqrt{4\mathrm{Dt}}}),$$ where the erf(y) is the error function, and the O_2_ concentration in the water is approximately equal to 0.044 mol/L [20]. According to the diffusion layer thickness formula δ = (πDt)^0.5^, the calculated boundary layer thickness when O diffuses for 1 s is 0.08 mm. Therefore, three positions near the wall—that is, y = 0.05 mm, diffusion layer interface y = 0.1 mm, and outside the diffusion layer y = 0.15 mm—are taken to calculate the change in O concentration C^y^. As shown in Figure 13, when the wall surface O concentration was 1/10 of the diffusion outer layer concentration, the O molecules participating in the electrochemical cathode reaction accounted for 90% of the total molecules. The O concentration in the diffusion layer (x = 0.05 mm) significantly decreases with time—that is, approximately 38% in 1 s. Meanwhile, the diffusion layer interface (x = 0.1 mm) only decreases by 10%. This result shows that in the film, within a short time of the formation of the electrochemical system, the concentration of the reactants forms a larger concentration gradient in the diffusion layer because the oxygen content that is involved in the reaction accounts for 90%, the electrochemical reaction is rapid, and the system is controlled by the reactant concentration.

According to the solubility of O_2_ and CO_2_ in water at different temperatures tested by Duan and Broden [20,21], as shown in Table 3, the solubility of CO_2_ in water is 10- to 20-fold higher than that of O_2_ at temperatures ranging from 50 to 200 °C. This result shows that CO_2_ derived more reactants than O_2_ in a high-temperature environment. However, under the coexistence of O_2_ and CO_2_, O_2_ first combines with Fe^2+^ or Fe^3+^ to form Fe(OH)_3_ (Equations (8) and (9)) because O_2_ oxidizes more than H_2_CO_3_ and H^+^. A part of Fe^2+^ can combine with HCO^3−^ to form FeCO_3_ crystals (Equation (10)). In a high-temperature environment, Fe(OH)_3_ and FeCO_3_ are decomposed into Fe_2_O_3_, which has poor adhesion and poor ability to form a film, as shown in Equations (11)–(13).
(8)$$\mathrm{First reaction stage}\;{4\mathrm{Fe}}^{2+}{+4H}^{+}{+O}_{2}\to {4\mathrm{Fe}}^{3+}{+2H}_{2}O,$$
(9)$$\mathrm{First reaction stage}\;{\mathrm{Fe}}^{3+}{+3H}_{2}O\to {\mathrm{Fe}(\mathrm{OH})}_{3}{+3H}^{+},$$
(10)$$\mathrm{First reaction stage}\;{\mathrm{Fe}}^{2+}{+\mathrm{HCO}}_{3}^{-}\to {\mathrm{FeCO}}_{3}{+H}^{+}+2e,$$
(11)$$\mathrm{Second reaction stage}\;{\mathrm{Fe}(\mathrm{OH})}_{3}\to {\mathrm{FeO}(\mathrm{OH})+H}_{2}O,$$
(12)$$\mathrm{Second reaction stage}\;2\mathrm{FeO}(\mathrm{OH})\to {\mathrm{Fe}}_{2}O_{3}{+H}_{2}O,$$
(13)$$\mathrm{Second reaction stage}\;{4\mathrm{FeCO}}_{3}{+O}_{2}\to {2\mathrm{Fe}}_{2}O_{3}{+4\mathrm{CO}}_{2}.$$

In addition to the effect on the priority of the reaction on the metal surface, the penetration of O_2_ into the product film will also affect the surface structure. As shown in Figure 14, local reactions occurred inside the product film because the FeCO_3_ film was not enough to isolate O_2_ molecules, and the liquid flow made the product film thin. When O_2_ molecules enter the film, they will react with FeCO_3_ to form Fe_2_O_3_ and CO_2_, and a small amount of H_2_ will be generated by the partial H evolution reaction. When gas accumulates inside, it will swell inside the product film and eventually break the film. The flow of external liquid will also accelerate the rupture of the corrosion film.

### 4.3. Influence of Temperature

In addition to the influence of mass transfer, reaction priority, and reaction position on the corrosion of N80 in the coexisting environment, temperature also changes the progress and rate of several reactions. First, temperature will affect the corrosion process dominated by chemical reactions and the dissolution and diffusion of CO_2_/O_2_ in the solution. Second, temperature will affect the oxidation of Fe^2+^, the deposition of corrosion product FeCO_3_, and the formation of corrosion intermediate products. These factors increase the complexity of the corrosion process and surface characteristics of the material.

Temperature can affect the gas dissolution and diffusion rate and further affect the composition, compactness, and corrosion rate of the corrosion product film of N80 steel in the CO_2_–O_2_ environment. As shown in Figure 2, when the partial pressures of CO_2_ are 0.50 and 0.75 MPa, and the temperature is less than 50 °C, the slow diffusion rate of O_2_ causes the surface corrosion of the material to be dominated by CO_2_ corrosion. When the temperature continues to increase and is less than 150 °C, part of the FeCO_3_ film will be transformed into Fe_2_O_3_. At this time, the corrosion rate increases because Fe_2_O_3_ is loose and easily falls off. Meanwhile, due to changes in the structure of the corrosion product film, a part of the corrosion product film is easily penetrated by O_2_ and Cl^−^. Fe ions can also diffuse into the solution through the microporous structure on the corrosion product film, which accelerates the dissolution of iron. When the temperature is higher than 150 °C, several iron oxides become dense and protective, and the corrosion rate will decrease.

## 5. Conclusions

To study the corrosion law of tubing materials in HPHT flowing media, corrosion experiments were carried out in a test liquid containing O_2_ and CO_2_. According to the corrosion rate, corrosion morphology, and product composition obtained by the experiments, the following conclusions can be drawn.

(1) When the CO_2_ concentration was low in the coexistence environment, the corrosion rate of N80 was affected by O_2_ and continued to increase with increasing temperature. When the CO_2_ concentration increased (when the volume fraction was higher than 10%), the corrosion rate had a maximum value near 100 °C.

(2) O_2_ and temperature affected the changes in intermediate products and film structure of the CO_2_ corrosion. The oxide formed by the reaction of O_2_ and Fe will cover the surface of regular FeCO_3_ grains, which may destroy the original single structure of the product film and cause the surface film to rupture or form cracks. The temperature increase will also accelerate the rupture process.

## Figures and Tables

**Figure 1 materials-13-05214-f001:**
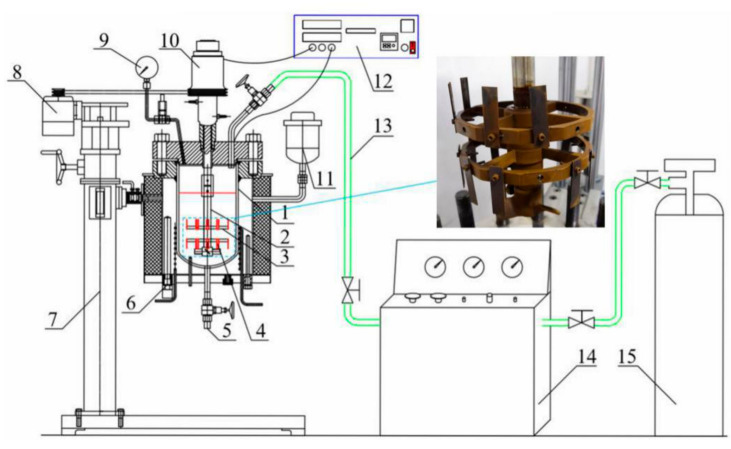
Schematic of experimental setup. 1. Autoclave body; 2. Agitator shaft; 3. Sample fixture; 4. Sample; 5. Outlet; 6. Electric heater; 7. Lifting device; 8. Motor; 9. Pressure gauge; 10. Magnetic stirrer; 11. Expander; 12. Control cabinet; 13. Gas pipeline; 14. Booster pump; 15. CO_2_ storage tank.

**Figure 2 materials-13-05214-f002:**
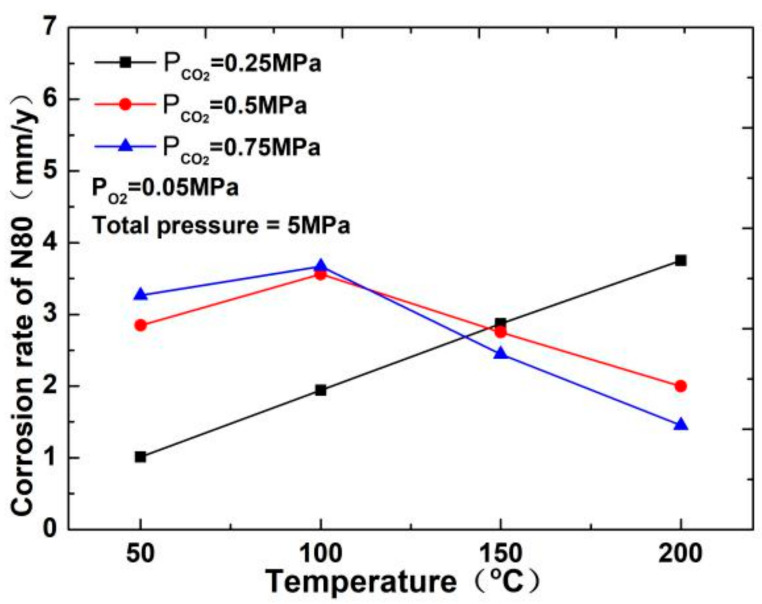
Corrosion rate of N80 steel at different temperatures.

**Figure 3 materials-13-05214-f003:**
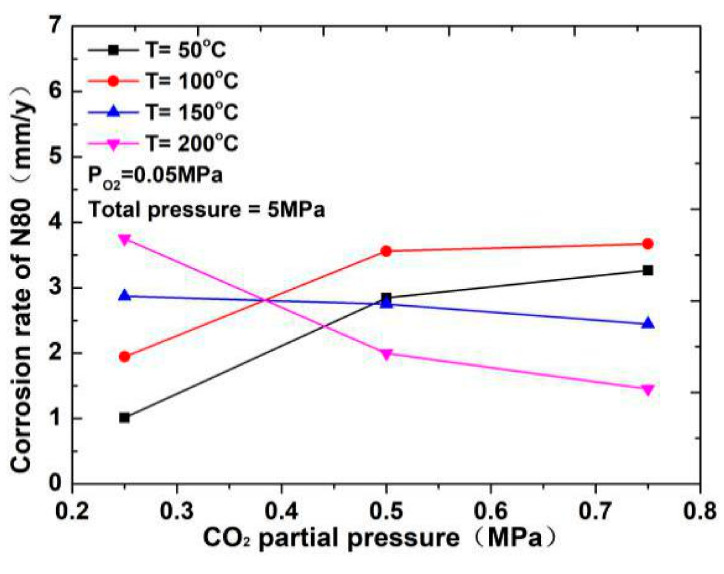
Corrosion rate of N80 steel at different CO_2_ partial pressures.

**Figure 4 materials-13-05214-f004:**
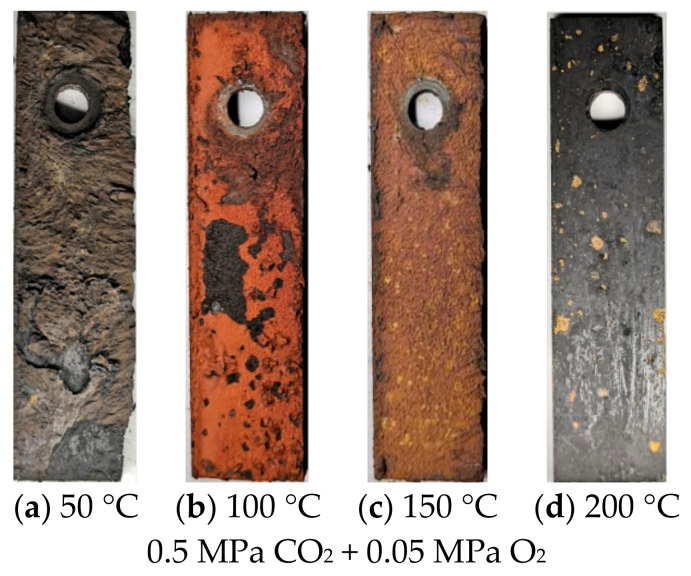
Corrosion macroscopic morphology of N80 steel at different temperatures in O_2_ and CO_2_ environment.

**Figure 5 materials-13-05214-f005:**
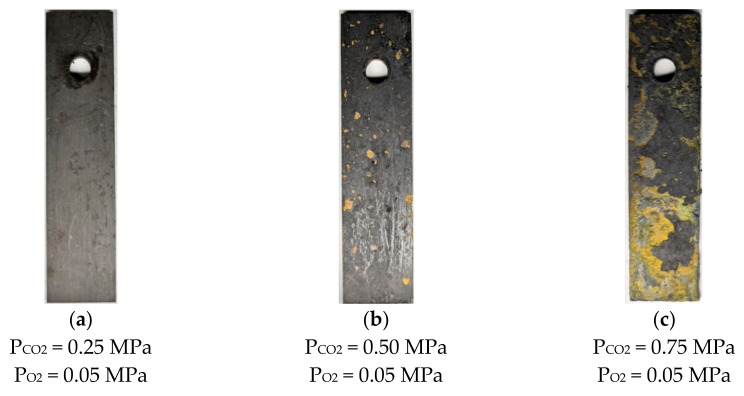
Corrosion macroscopic morphology of N80 steel at different CO_2_ partial pressures (200 °C).

**Figure 6 materials-13-05214-f006:**
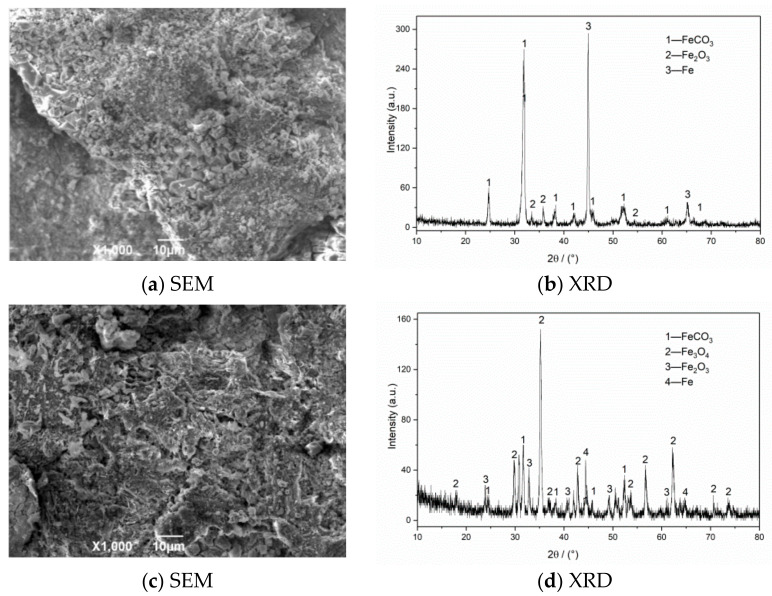
SEM morphology and XRD of N80 steel under different temperatures in environment of coexisting CO_2_–O_2_, (**a**,**b**) T = 50 °C, (**c**,**d**) T = 100 °C, (**e**,**f**) T = 150 °C, (**g**,**h**) T = 200 °C. P_CO2_ = 0.5 MPa, P_O2_ = 0.05 MPa.

**Figure 7 materials-13-05214-f007:**
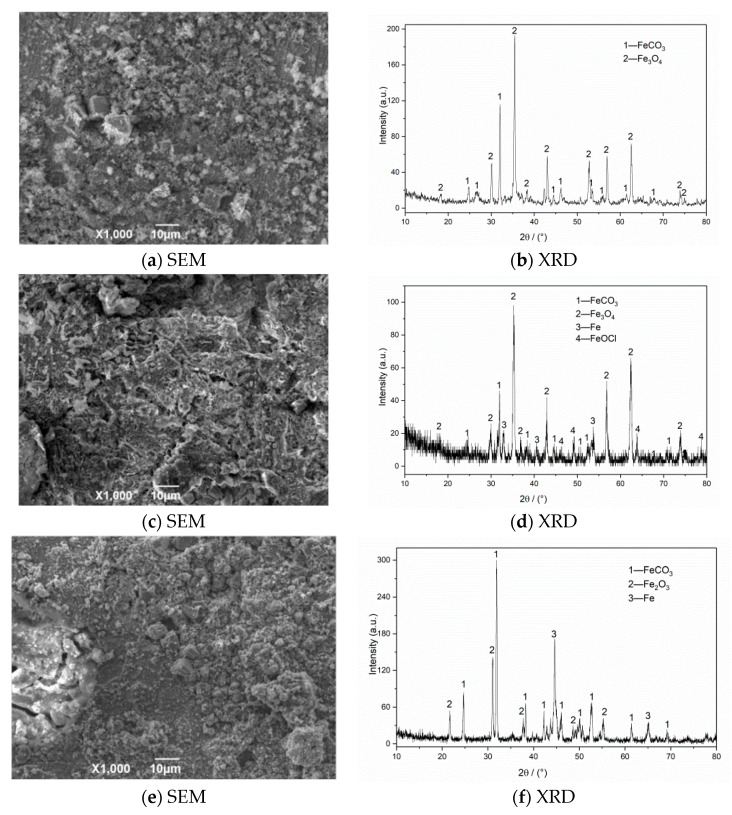
SEM morphology and XRD of N80 steel under different CO_2_ partial pressures in environment of coexisting CO_2–_O_2_; (**a**,**b**) P_CO2_ = 0.25 MPa, (**c**,**d**) P_CO2_ = 0.50 MPa, (**e**,**f**) P_CO2_ = 0.75 MPa. P_O2_ = 0.05 MPa, T = 100 °C.

**Figure 8 materials-13-05214-f008:**
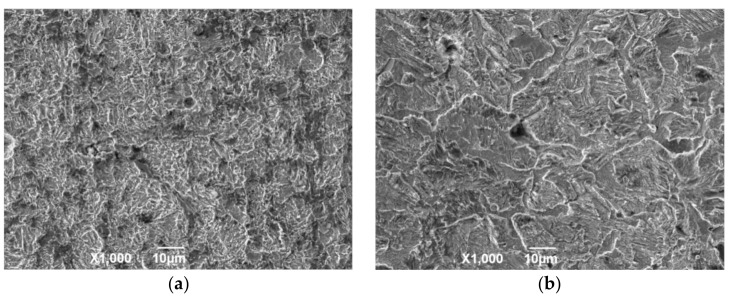
SEM morphology of N80 steel after removing corrosion product film; (**a**) T = 50 °C, (**b**) T = 100 °C, (**c**) T = 150 °C, (**d**) T = 200 °C, P_CO2_ = 0.50 MPa, and P_O2_ = 0.05 MPa.

**Figure 9 materials-13-05214-f009:**
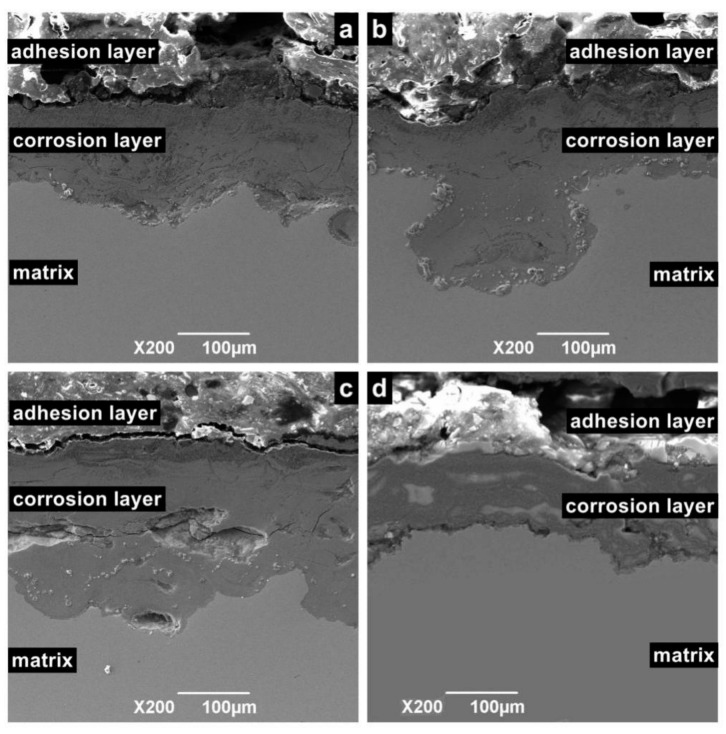
Cross-sectional SEM morphology of N80 steel at different temperatures in environment of coexisting CO_2_–O_2_; (**a**) T = 50 °C, (**b**) T = 100 °C, (**c**) T = 150 °C, (**d**) T = 200 °C, P_CO2_ = 0.50 MPa, and P_O2_ =0.05 MPa.

**Figure 10 materials-13-05214-f010:**
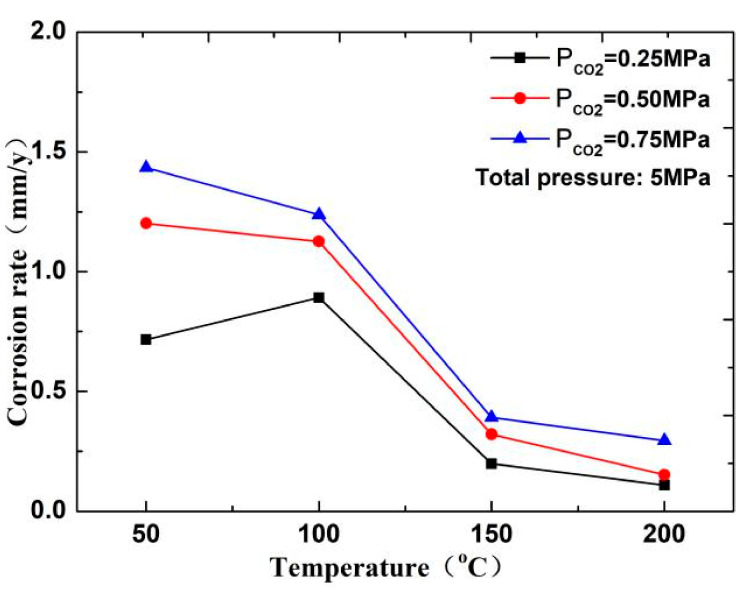
Corrosion rate of N80 steel at different temperatures in CO_2_ environment.

**Figure 11 materials-13-05214-f011:**
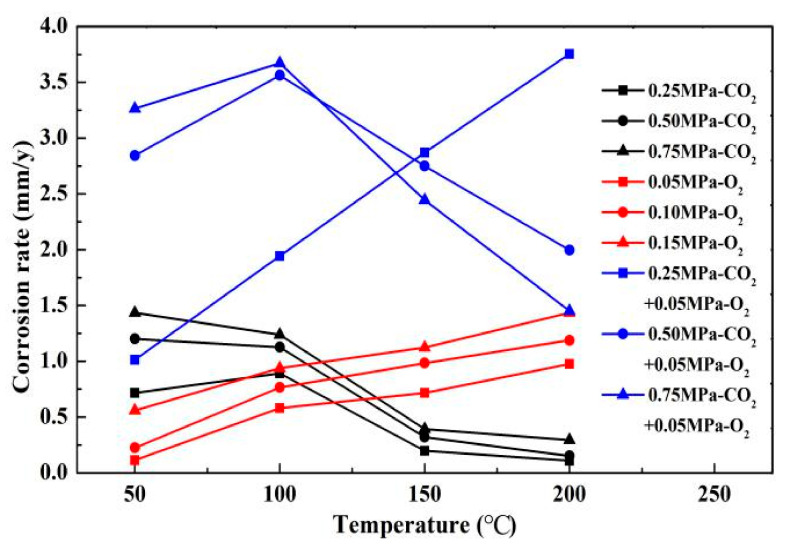
Comparison of corrosion rates of N80 steel in different media.

**Figure 12 materials-13-05214-f012:**
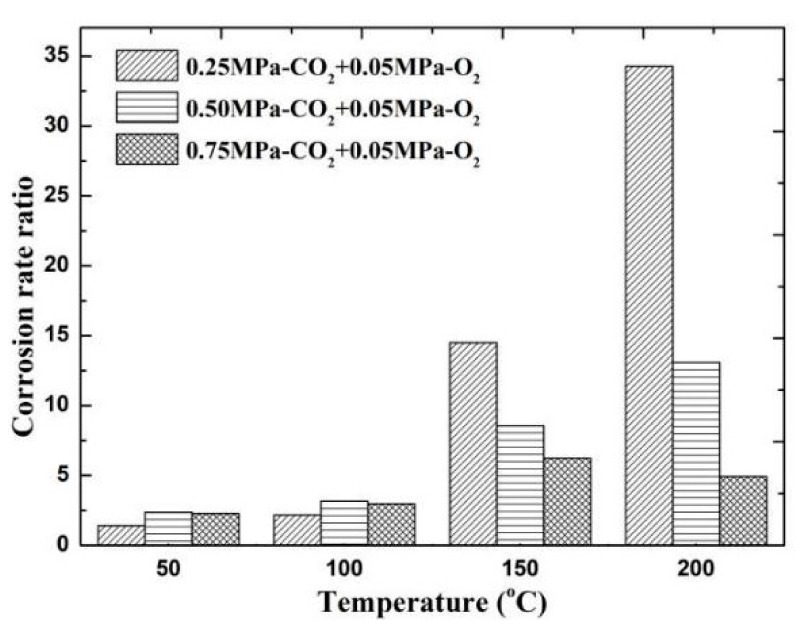
Comparison of corrosion rates of O_2_ and O_2_–CO_2_ coexistence conditions.

**Figure 13 materials-13-05214-f013:**
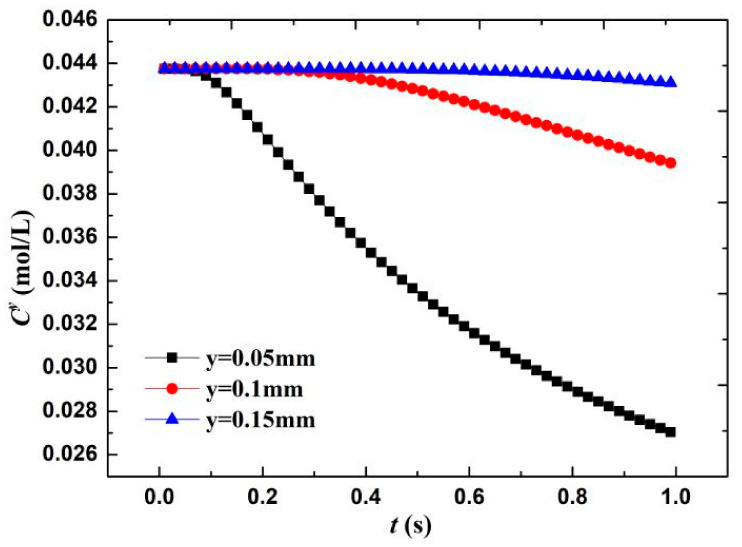
Change in O_2_ concentration on material surface with reaction time.

**Figure 14 materials-13-05214-f014:**
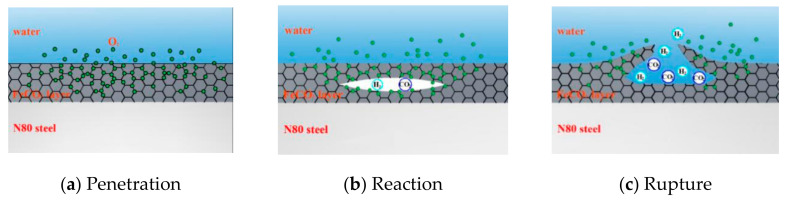
Schematic of corrosion in high-temperature environment of coexisting O_2_ and CO_2._

**Table 1 materials-13-05214-t001:** Chemical composition of N80 steel (wt%).

Materials	C	Si	Mn	P	S	Cr	Mo	Ni	Cu	Fe
N80	0.22	0.21	1.77	0.01	0.003	0.036	0.021	0.028	0.019	97.683

**Table 2 materials-13-05214-t002:** Composition of test solution used in corrosion experiments.

HCO_3_^−^ (mg/L)	Cl^−^ (mg/L)	Ca^2+^ (mg/L)	Na^+^ (mg/L)	SO_4_^2−^ (mg/L)	Salinity (mg/L)	pH
800	2000	10	1785	400	4995	7.8

**Table 3 materials-13-05214-t003:** Solubility of O_2_ and CO_2_ in water under 5 MPa pressure (mol/L) [20,21].

	T (°C)	50	60	90	100	120	130	150	210
Gas									
O_2_		0.044	—	—	0.036	—	0.038	0.044	—
CO_2_		—	0.669	0.495	—	0.415	—	0.377	0.299

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
