# Peer review of "Experimental Study on Corrosion Performance of Oil Tubing Steel in HPHT Flowing Media Containing O2 and CO2"

_materials, 2020, doi:10.3390/ma13225214_

Round 1

Reviewer 1 Report

The authors prepared an interesting research paper on a novel technique for tubing materials testing at high temperatures in liquid environments containing gaseous elements. The quality of experimental works is sufficient, however, there are some important and questionable areas that need to be clarified or corrected.

  1. Page 1, paragraph 1: Authors claim: “The results show that temperature and gas concentration are the main factors affecting material corrosion in coexisting environments.” This is a little bit misleading statement as there is a lack of other experiments testing other factors influencing materials corrosion in this study (for example: electrolyte composition, the velocity of stirring, material composition,…). Hence, the sentence needs to be re-stylized.
  2. Page 1, paragraph 1: “Meanwhile, the flowing…” instead of “Meanwhile, The flowing…”
  3. Page 1, paragraph 1 of introduction: This paragraph is not supported by references. Need to be added.
  4. Page 2, paragraph 1: I could not find any information from the sentence: ”Although the carbonic acid is a kind of dibasic weak acid, the acidity is higher than that of hydrochloric acid under the same pH value, which will cause severe corrosion to the material [1]” in the reference [1]. Could you explain it, please?
  5. Page 2, paragraph 1: Could you explain the statement, that the acidity of carbonic acid is higher than that of HCL under the same pH value? Do you claim, that the same pH does not mean the same acidity? I am a little bit confused about this sentence. (”Although the carbonic acid is a kind of dibasic weak acid, the acidity is higher than that of hydrochloric acid under the same pH value, which will cause severe corrosion to the material [1]”)
  6. Page 2, paragraph 1: Authors claim: ”… galvanic corrosion between corrosion products and reaction solution…” I did not hear about galvanic corrosion between metal and liquid. Galvanic corrosion is attributed to contact of different metals or different phases within the same material (micro-galvanic corrosion). You have to explain it.
  7. Page 3, paragraph 2: The sentence: “HPHT, complex and flowing media were the environment that N80 tubing string has to be corroded experience during the service period.” needs to be re-stylized as it does not make sense.
  8. Page 3, paragraph 1 of the experimental system: “…can resist corrosion under…” instead of “…can resistant corrosion under…”
  9. Page 4, Table 1: Chemical composition is not complete. Fe is missing.
  10. Page 4, Table 2: Were the experiments performed in “produced water” or “simulated produced fluid in oilfield”? Please, unify the terminology. Why weren’t the experiments performed in a kind of oil, as the area of research is within the oilfield industry using oil tubing steel?
  11. Page 5, paragraph 1 of “macroscopic morphology…”: Use “…in an O2 and CO2 containing environment…” instead of “…in an O2 and CO2 environment...“.
  12. Page 5, paragraph 1 of “macroscopic morphology…”: Authors claim: “The results show that there are thicker corrosion products attached to the surface of the corroded material, which causes the metal matrix to be completely covered by precipitated scales.” The sentence is not complete. Thicker corrosion products than what, where?
  13. Page 11, Figure 9: The explanation of individual figures a) to D) is missing under the figure 9.
  14. Page 11: The sentence: “The essence of the corrosion of CO2 and steel materials is the electrochemical reaction of metals in carbonic acid solution.” needs to be re-stylized. The corrosion of CO2 does not exist.
  15. paragraph 1 of “Conclusion”: The sentence: “In order to study the corrosion law of tubing materials in HPHT flowing media, corrosion experiments in a single gas and O2 and CO2 coexisting gas environment were carried out.” needs to be re-stylized. The experiments were not carried out in a gas environment.

Author Response

Reply: Thank you for your meaningful suggestions. We have modified most of the questions and marked them in red font. We have asked native English speakers to proofread the language.

Reviewer 2 Report

  1. Table 1 needs 'Fe's balance'.
  2. Table 2; test solution instead of produced water
  3. 3.1 Effect of solution's temperature and CO2 partial pressure on the corrosion rate
  4. The unit of corrosion rate is mm/y than mm/a.
  5. Every figure's title contains the detail explanation- ex) (a), (b) etc. not below the photos.
  6. Which difference between FIg. 10 and Fig. 2?
  7. Which difference between Fig. 11 and Fig. 3?
  8. Fig.12; What is 'corrosion rate ratio'?
  9. Fig.13; What means y?
  10. Conclusions; Did you present the effect of flow velocity in this manuscript? If not, please delete and rewrite the conclusions.

Author Response

Reply: Thank you for your meaningful suggestions. We have modified most of the questions and marked them in green font. We have asked native English speakers to proofread the language.

Reviewer 3 Report

This paper attempts to report an experimental corrosion study of Oil Tubing Steel in HPHT Flowing Media Containing O2 and CO2. This research has the potential for publishing after improving the structural and scientific comments.

The following are some specific comments and points for consideration:

  1. Some sentences in the introduction need to certain references, including:

1.1. “As the only channel for oil or gas exploitation, the downhole tubing string is inevitably corroded
by the high pressure, high temperature and complex flowing solution, which may cause
deformation and fracture damage of tubing string, thereby affecting well site safety”.

1.2. “N80 tubing steel, which is commonly used in thermal recovery wells, is damaged by the
corrosion of O2, CO2, and H2S in the liquid containing HCO3-, Cl-, SO42-, Ca2+, Mg2+, and Na+”.

1.3. “The essence of the CO2 corrosion for tubing steel is the electrochemical corrosion of metals in aqueous carbonate solution”

1.4. “The results showed that the corrosion rate of steel in CO2 solution is controlled by hydrogen evolution kinetics”.

1.5. “In the presence of O2, there is a double-layer film structure in the CO2 corrosive system, the inner
is mainly Fe2O3, and the outer is the loose and porous corrosion product FeCO3”.

1.6. “The corrosion product film on the surface of N80 steel was denser and the adhesion was higher at a lower Cl content (5000 mg/L), which indicated the corrosion resistance was better”.

1.7. “As a result, the deposition tendency of CaCO3 increases”.

1.8. “Therefore, as the Cl- content is increased to 100000 mg/L, the corrosive film with high hardness and high adhesion is deposited on the tubing steel surface, which improves the erosion resistance of N80”.

1.9. “This is because when the corrosive environment contains both CO2 and O2, FeCO3 and
Fe(OH)2 corrosion product films will appear on the initial surface, and then O2 in the environment
will oxidize with Fe(OH)2 and FeCO3 to form iron oxidation”.

  1. In this phrase, “In addition, because the corrosion products on the steel surface are not covered uniformly at different temperatures, it may cause the formation of galvanic corrosion between corrosion
    products and steel surface and the galvanic corrosion between corrosion products and reaction
    solution, which accelerates the CO2 corrosion”. Authors must clearly describe the galvanic coupling mechanism due to galvanic corrosion between corrosion products and steel surface in the introduction section”.
  2. Authors should describe the XRD sample preparing procedure and also information about the XRD analysis process in the experimental section”.
  3. It is better to merge the Fig.3 to Fig.2b.
  4. There is no correlation between the corrosion rate of N80 at 200 °C (versus CO2 partial pressure) in Fig.3 and optical images in Fig.5. Authors must describe this inverse correlation.
  5. Did you capture the SEM images in Fig.7 at 100 or 200 °C???, because it doesn’t make a correlation with optical images in Fig.5.
  6. In Fig.8, authors must clearly explain these pitting corrosions were generated or initiated because of inclusion or Not ???. It is important to consider that N80 steel has a 1.7% Mn and 0.003% S, so pitting initiation in inclusions is more susceptible.
  7. The authors should mark the corrosion pits in Fig.8 with arrows.
  8. In the figure caption and also SEM images in Fig.9, authors should describe and distinguish the different temperatures.
  9. Also, authors should consider the references for supporting some scientific sentences in the results and discussion section. It is an un-meaning work that using scientific sentences from previous studies without any referees.

Author Response

Reply: Thank you for your meaningful suggestions. We have modified most of the questions and marked them in blue font. We have asked native English speakers to proofread the language.
